# SCHRODINGER'S MEMORY: LARGE LANGUAGE MODELS

## ABSTRACT

Memory is the foundation of all human activities; without memory, it would be nearly impossible for people to perform any task in daily life. With the development of Large Language Models (LLMs), their language capabilities are becoming increasingly comparable to those of humans. But do LLMs have memory? Based on current practice, LLMs do appear to exhibit memory. So, what is the underlying mechanism of this memory? Previous research lacked a deep exploration of LLMs' memory capabilities and the underlying theory. In this paper, we use the Universal Approximation Theorem (UAT) to explain the memory mechanism in LLMs. We also conduct experiments to verify the memory capabilities of various LLMs, proposing a new method to assess their abilities based on the memory ability. We argue that LLM memory operates like Schrödinger's memory, meaning that it only becomes observable when a specific memory is queried. We can only determine if the model retains a memory based on its output in response to the query; otherwise, it remains indeterminate. Finally, we expand on this concept by comparing the memory capabilities of the human brain and LLMs, highlighting the similarities and differences in their operational mechanisms.

## 1 INTRODUCTION

Language is not only one of humanity's most important abilities but also the foundation of communication (Miller, 1951), knowledge transfer (Han & Ellis, 1998), and the development of civilization (Yu, 2015). Language models can be seen as simulations of human intelligence, enabling them to perform tasks traditionally achievable only by humans. Currently, LLMs based on the Transformer architecture have become one of the hottest topics in artificial intelligence research today. These models have acquired some human-like language capabilities and are already impacting daily life in areas such as machine translation (Brants et al., 2007; Moslem et al., 2023), text summarization (Van Veen et al., 2024; Zhang et al., 2019), sentiment analysis (Zhang et al., 2023a; Mao et al., 2022; Zhang et al., 2023b), question-answering systems (Masry et al., 2022; Xu et al., 2023), and text generation (Bai et al., 2023; Yang et al., 2024a; OpenAI et al., 2024).

Although the performance of LLMs is impressive, research on their memory mechanisms remains limited. Memory is a crucial capability for humans; without it, we would struggle to complete even the simplest tasks. For instance, in everyday conversations, we need to remember what others have said in order to respond appropriately, and this memory capacity facilitates smooth dialogue. Memory plays a vital role in guiding various aspects of our daily lives. As LLMs become increasingly powerful, an important question arises: do these models possess memory? If so, in what form does it exist, and how does it differ from human memory? Current research on LLM memory primarily focuses on two main directions:

Expanding Context Length: This approach aims to equip LLMs with more memory by extending the context window (Chen et al., 2023; Zhu et al., 2023; Yang, 2023; Fei et al., 2023). Since short contexts fail to provide enough information, increasing the context length allows the model to maintain more comprehensive information across long sequences.

External Memory Integration: This method involves building memory storage systems (Graves et al., 2014; Xiao et al., 2024; Wu et al., 2022; Yang et al., 2024b) that encode and store past events (Zhang et al., 2023c), allowing the model to retrieve and update memories on disks as needed. Such mechanisms enable models to forget or reinforce certain memories over time.

Although these studies have made progress in addressing the memory limitations of LLMs, they have not fully explained how memory functions within these models. For example, when asked, "Who is the President of the United States?" LLMs like GPT-4 (OpenAI et al., 2024) or Llama-3 (Dubey et al., 2024) may say "Trump". It is outdated information, but it also indicates that some form of memory is indeed present in LLMs. However, this memory does not come from an external storage unit but is inferred by the model based on the input. The articles Jagielski et al. (2022); Carlini et al. (2022) attempt to evaluate the memory capabilities of LLMs. However, due to the lack of a fundamental theoretical framework, the definition of memory itself remains vague, resulting in conclusions that are largely based on straightforward experimental observations. This raises fundamental questions: Why do LLMs exhibit this ability to infer previously learned information from the input? How does this differ from human memory? Where is memory stored in LLMs?

In this paper, we use UAT theory to explain this ability of recalling information learned from the past based on input cues. We argue that this information can be understood as a dynamic approximation capability of UAT (Wang & Li, 2024b), where the model fits a corresponding result based on the input, and the observed phenomenon appears as memory. We call this "Schrödinger's memory" because we can only determine whether the LLMs have this memory by asking it and analyzing its response; otherwise, the memory remains indeterminate. Additionally, we evaluate the memory capabilities of several models and propose that this approach can be used to assess the overall ability of LLMs. The contributions of this work are as follows:

- We explain LLMs' memory abilities through the lens of UAT.
- We propose a new, objective method for evaluating LLMs' capabilities: memory ability assessment.
- We logically make a comparison between the memory of LLMs and human memory and reasoning capabilities.

The structure of this paper is as follows: In Section 2, we briefly explain the UAT and present mathematical formulation of multi-layer Transformers in the form of UAT. In Section 3, we provide both theoretical and experimental evidence demonstrating the memory capabilities of LLMs. Finally, in Section 4, we conduct a comprehensive analysis of human and LLM abilities, with a focus on memory ability.

## 2 UAT AND LLMS

The UAT (Cybenko, 2007; Popescu et al., 2009) serves as the foundational theory of deep learning. Our goal is to theoretically explain memory of Transformer-based LLMs using the UAT framework. To do this, we will first present the mathematical form of UAT in Section 2.1, followed by the corresponding UAT form for LLMs in Section 2.2. We will then use this UAT form to explain the memory abilities of LLMs.

### 2.1 UAT

In this section, we provide a brief overview of the UAT, which was first proposed by Cybenko (2007). As stated in Theorem 2 by Cybenko (2007), if $\sigma$ represents any continuous sigmoidal function, then a finite sum of the following form:

$$G(\mathbf{x}) = \sum_{j=1}^{N} \alpha_j \sigma \left( \mathbf{W}_j^{\mathrm{T}} \mathbf{x} + \theta_j \right) \tag{1}$$

is dense in $C\left(\mathbf{I}_n\right)$. Here, $\mathbf{W}_j \in \mathbb{R}^n$ and $\alpha_j, \theta \in \mathbb{R}$ are fixed. For any $f \in C\left(\mathbf{I}_n\right)$ and $\varepsilon > 0$, there exists a function $G(\mathbf{x})$:

$$|G(\mathbf{x}) - f(\mathbf{x})| < \varepsilon \quad \text{for all} \quad \mathbf{x} \in \mathbf{I}_n. \tag{2}$$

This suggests that, for a sufficiently large $N$, a neural network can approximate any continuous function on a closed interval. Hornik et al. (1989) further demonstrates that multilayer feedforward

networks conform to the UAT, enabling them to approximate arbitrary Borel measurable functions. In the context of Eq. (1), where the function $G(\mathbf{x})$ produces a scalar in $\mathbb{R}$, this framework naturally generalizes when $G(\mathbf{x})$ maps to $\mathbb{R}^m$, requiring approximation for each dimension. To accommodate this multidimensional output, a simple adjustment to Equation (1) is needed: the transformation matrix $\mathbf{W}_j$ is modified to reside in $\mathbb{R}^{n \times m}$, the bias term $\theta_j$ is redefined as a vector in $\mathbb{R}^m$, and $\alpha_j$ is reshaped into a matrix.

## 2.2 THE UAT FORMAT OF TRANSFORMER-BASED LLMS

Current LLMs are primarily based on Transformer architecture. In UAT2LLMs (Wang & Li, 2024b), it has already been demonstrated that the mathematical structure of multi-layer Transformers aligns with the UAT in a general sense. However, unlike the original UAT, the UAT form of Transformer-based models has the ability to dynamically fit functions based on the input. Figure 1 illustrates a basic module in Transformer, and according to UAT2LLMs, the corresponding UAT form for Figure 1 is:

$$\mathbf{x}_{i+1} = (\mathbf{W}'_{i+1,1}\mathbf{x}_0 + \mathbf{b}_{i+1,1}) + \sum_{j=1}^{i+1} \mathbf{W}'_{j,3}\sigma(\mathbf{W}'_{j,2}\mathbf{x}'_0 + \mathbf{b}'_{j,2}) \tag{3}$$

where $\mathbf{x}_{i+1}$ represents the output of the $i+1$-th layer, with $\mathbf{x}_0$ as the network's input. The term $\mathbf{b}'_{j,2}$ is computed as $(\mathbf{W}'_{j,2}\mathbf{b}'_{j-1,3} + \mathbf{b}'_{j,2}) + \mathbf{W}'_{j,2}UAT^R_{j-1}$, where $UAT^R_{j-1} = \sum_{k=1}^{j-1} \mathbf{W}'_{k,3}\sigma(\mathbf{W}'_{k,2}\mathbf{x}'_0 + \mathbf{b}'_{k,2})$. The value of $\mathbf{b}'_{j,2}$ is approximated by the $j$-th layer of the UAT, with $\mathbf{x}_0$ as the input. This allows the model to dynamically adjust functions based on the input. According to UAT2LLMs, parameters in the multi-head attention mechanism are modified dynamically in response to the input. Therefore, in the formula above, all parameters $\mathbf{W}'_{j,1}$, $\mathbf{W}'_{j,2}$, and $\mathbf{W}'_{j,3}$ in layer $i$, where $j = 1, \ldots, i$, are dynamically adjusted based on the input.

Based on Eq. (3) and Eq. (1), it is clear that the multi-layer Transformer shares the same mathematical structure as the UAT. However, compared to the mathematical form of UAT in Eq. (1), the weights and bias parameters in Eq. (3) can dynamically change according to the input. This ability enables the Transformer to adaptively fit based on the input, whereas the UAT's parameters are fixed once training is completed, limiting it to fitting static functions and rendering it incapable of responding to dynamic changes in input data. This dynamic fitting capability is the ultimate source of the powerful memory observed in LLMs.

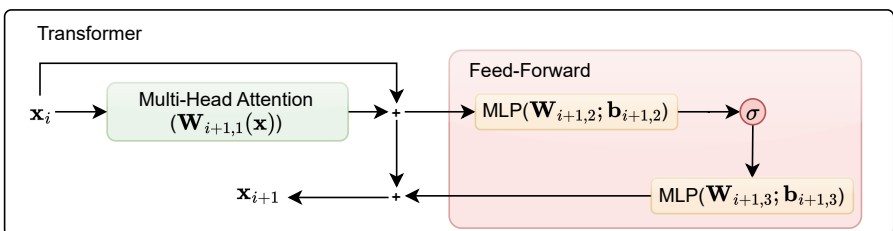

Figure 1: The basic block in Transformer-based LLMs.

# 3 THE MEMORY OF LLMS

In this section, we will demonstrate the memory capabilities of LLMs. First, in Section 3.1, we provide a clear definition of memory. Then, in Section 3.2 and 3.3, we give a discription to datasets and explain the memory mechanism of LLMs using UAT theory and validate their memory characteristics through experiments. In Section 3.4, we explore the impact of input length on the accuracy of LLM memory.

## 3.1 THE DEFINITION OF MEMORY

Before delving into the study of memory in LLMs, it is important to first define or provide a relatively precise description of what memory is. According to Wikipedia:

- Memory is the faculty of the mind by which data or information is encoded, stored, and retrieved when needed.

However, this definition has some fundamental issues. Encoding data or information is not problematic, as information in the brain is transmitted via electrical signals, and we need to encode that information in a way the brain can process. The problem arises with the concepts of "storage" and "retrieval." The brain does not have a structure analogous to a database for storing information. So, where is this information actually stored? Is it in the neurons of the brain? If so, does a single neuron store a word, or does it store an entire sentence? Next, we give an exmple:

| Question 1: | What is Newton's first law? |
| --- | --- |
| Answer 1: | Every object perseveres in its state of rest, or of uniform motion in a right line, except insofar as it is compelled to change that state by forces impressed thereon. |

So, is this sentence stored within a single neuron? Or does each neuron store just a word, with a specific region of the brain dedicated to this particular memory? Given the vast amount of information humans receive daily, can neurons truly store such an immense volume of data without hindering normal cognitive processes? After all, almost every routine activity requires memory. Take, for example, the simple task of going to the cafeteria: we need to remember when to go, the cafeteria's location, the route to get there, which foods are available, what counts as utensils, where to find them, and how to use them.

Moreover, if this memory is stored in a fixed set of neurons, then every time the question is raised, the response should be identical, since the retrieval would be from the same static content. Every word in the response should be exact, with no omissions or additions (even if the information has been abstractly encoded, as long as the encoding and decoding processes are consistent, the content should remain unchanged). This, however, is clearly unreasonable. Therefore, we provide a more precise definition of the concept of "memory":

Memory is defined by two key components: input and output.

- **Input:** To trigger a memory, the input must be the same or similar to information that the brain (or LLM) has previously encountered.

- **Output:** The result is based on the input, which could be correct, incorrect, or forgotten. If the result is correct, it means it aligns with information previously acquired.

We need to stress that a key requirement for recalling a memory is the presence of input—without input, there is no memory, as memory is activated by input. Even if the brain holds a memory of something, it cannot be determined whether that memory exists unless it is prompted by a specific query. Without specific input conditions, a person wouldn't recall a particular event.

Using Question 1 as an example, the input is: "What is Newton's first law?" Without this input, no one would suddenly recall Newton's first law. The recollection of Newton's first law is triggered by input related to the theoretical context. This is why input is a necessary condition for memory, as it is the input that stimulates recall. The memory might be accurate, or it might be incorrect, indicating a deviation from previously acquired information—this deviation could be minor, significant, or even total forgetting. For example:

| | Question 1: | What is Newton's first law? |
| --- | --- | --- |
| Answer 2: | Minor distortions | Every object perseveres in its state of rest, except insofar as it is compelled to change that state by forces impressed thereon. |
| Answer 3: | Severe distortions | Every object always perseveres in its state of rest. |
| Answer 4: | Memory loss | I do not know. |

In summary, the term "memory" was traditionally used to refer specifically to human memory before the emergence of LLMs. Now, we believe that LLMs also exhibit memory. Therefore, we will verify the memory characteristics of LLMs based on the definition of memory outlined above.

## 3.2 DATASET

We utilized publicly available datasets from Hugging Face: CN Poems (Unknown, 2024a) for Chinese memory and ENG Poems (Unknown, 2024b) for English memory. We select the poems from datasets and the requirement is the combined length of the input and output to a maximum of 256 characters. Due to differences in character encoding between Chinese and English, a single Chinese character usually corresponds to one token, while an English word may map to multiple tokens. As a result, after tokenization, the length of Chinese input remains almost unchanged, with a maximum of 256 tokens. In contrast, the English input expands to a maximum of 730 tokens after tokenization. For the experiment, we selected 2,000 poems from each dataset.

## 3.3 THE MEMORY MECHANISM AND ABILITY OF LLMS

In Section 2.2, we have introduced the UAT format corresponding to Transformer-based LLMs. This UAT format can dynamically adjust to fit the corresponding output based on the input. Following this line of thought, we can also consider the memory of LLMs as being driven by inputs that fit specific outputs. In this context, the inputs consist of questions related to previously learned knowledge, while the outputs are responses based on that past knowledge. To explore this hypothesis, we designed a simple experiment.

We preprocessed the data in line with typical human memorization habits, allowing the LLMs to output the content of poems based on basic input information. For CN Poems, the input consisted of the dynasty, author, and title, while for ENG Poems, the input was the author and title. To test the memory ability of LLMs, we define the accuracy of memory as follows:

$$\text{Acc} = \frac{\Sigma_{i=1}^{N} 1_{Pred_i = True_i}}{N} \quad (4)$$

where $N$ is the number of examples, $Pred_i$ and $True_i$ are the prediction and ground true of the $i$-th exmple. We fine-tuned the CN Poems and ENG Poems on Qwen series models (Bai et al., 2023; Yang et al., 2024a) and bloom series models (Workshop et al., 2023) for 100 epochs. The results are shown in Table 1.

Table 1: The memory ability of Qwen1.5-0.5B-Chat, Qwen2-0.5B-Instruct, Qwen2-1.5B-Instruct, bloom-389m-zh, bloom-1b4-zh, bloom-560m, bloom-1b7 on CN Poems and ENG Poems.

| Models | | Qwen1.5 -0.5B-Chat | Qwen2 -0.5B-Instruct | Qwen2 -1.5B-Instruct | bloom -389m-zh | bloom -1b4-zh | bloom -560m | bloom -1b7 |
|---|---|---|---|---|---|---|---|---|
| CN Poems | Acc | 68.85 | 77.5 | **96.9** | 75.55 | 96.6 | - | - |
| ENG Poems | Acc | 99.85 | 99.85 | **99.9** | - | - | 99.2 | 99.15 |

Table 1 demonstrates that LLMs possess memory capabilities, which align precisely with the definition of memory we established. The training process is akin to giving a person 2,000 poems and asking him to memorize as many as possible, with each poem read up to 100 times. In the CN Poems dataset, the top-performing models were Qwen2-1.5B-Instruct and bloom-1b4-zh, which memorized 1,938 and 1,932 poems, respectively. In contrast, for the ENG Poems dataset, nearly all models were able to memorize all the poems.

These results are remarkable. An average person, without specific memory training, would struggle to remember 1,000 poems under similar conditions, whereas LLMs managed to retain almost all 2,000 poems. However, models like Qwen1.5-0.5B-Chat, Qwen2-0.5B-Instruct, and bloom-389m-zh performed comparatively weaker on the CN Poems dataset. We believe this is primarily due to two factors. First, insufficient pretraining led to relatively poorer language comprehension. For instance, while Qwen2-0.5B-Instruct and Qwen1.5-0.5B-Chat are the same model size, Qwen2-0.5B-Instruct outperformed Qwen1.5-0.5B-Chat because it was trained on better pretraining data, resulting in stronger language comprehension. This improved language understanding, in turn, enhances memory capabilities. Performance tests in the Qwen documentation (Bai et al., 2023; Yang et al., 2024a) also support this, as Qwen2-0.5B-Instruct consistently outperforms Qwen1.5-0.5B-Chat across various tasks. Similarly, individuals with better language comprehension tend to learn and recite poetry more efficiently. Compared to Chinese data, the larger and more comprehensive

English datasets help train models with stronger language skills, explaining why LLMs perform better on English poetry memorization. The second factor may be that Chinese is a more complex language, which smaller models struggle to learn effectively. Larger models like Qwen2-1.5B-Instruct and bloom-1b4-zh show significantly better performance, nearing that of the ENG Poems dataset, compared to smaller models like Qwen2-0.5B-Instruct and bloom-389m-zh.

Figure 2 shows the results of generating entire poems based on input information. As seen in the figure, after training, the models can successfully recite a complete poem using only the title and author. Figure 2 provides examples of prediction errors. Although incorrect, the outputs still maintain consistency with the poem's title and basic structure.

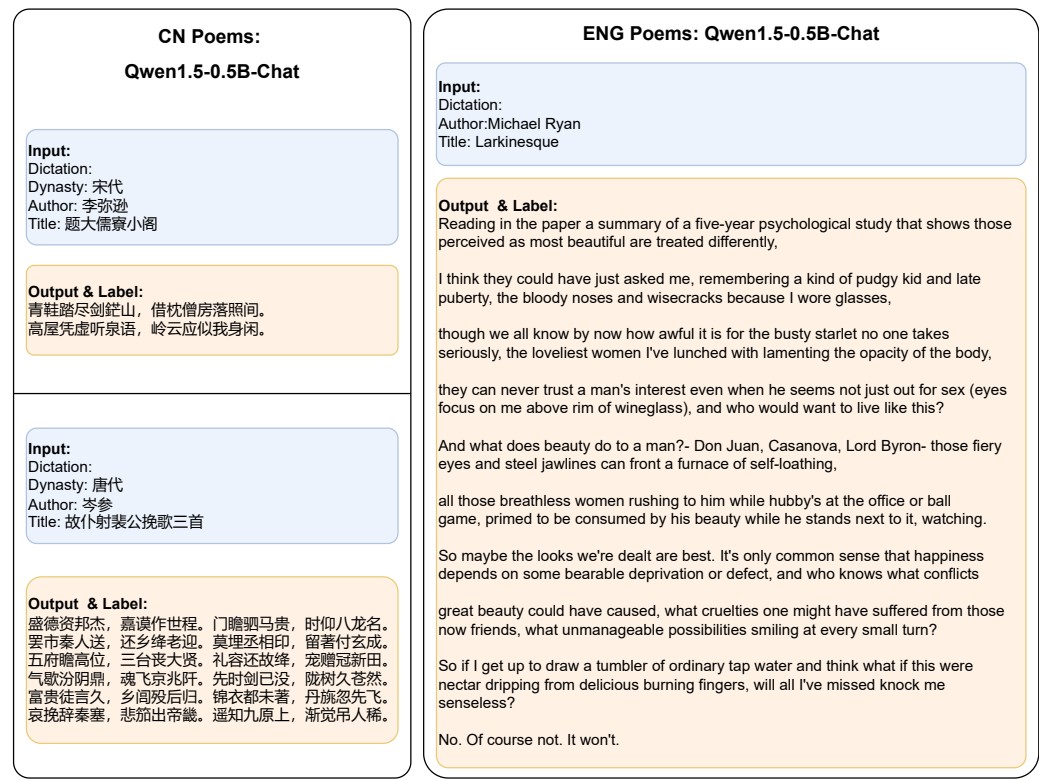

Figure 2: The examples of right predictions of CN Poems: Qwen1.5-0.5B-Chat and ENG Poems: Qwen1.5-0.5B-Chat which were fine-tuned separately on CN Poems and ENG Poems and subsequently tested the memory ability on their respective datasets, accurately recited the entire poem based on the input.

Based on these results, we believe that LLMs do indeed possess memory, and their memory mechanism works by fitting a specific output based on input. This is why we refer to LLMs' memory as "Schrödinger's memory"—we can only determine whether the LLMs have a particular memory when we ask a question and receive a response.

Moreover, we believe that memory capacity can also serve as an objective measure of LLMs' language abilities. Given the same training data, models of the same size, and the same number of training iterations, those which can retain more information generally exhibit stronger language skills. For example, in the case of Qwen1.5-0.5B-Chat and Qwen2-0.5B-Instruct, despite having the same model architecture, Qwen2-0.5B-Instruct demonstrates superior language ability due to differences in training sets, which in turn leads to better memory retention. This approach can also be used to assess the performance of models of different sizes. While it's known that larger models tend to have stronger memory capabilities, this method can help us roughly evaluate a model's upper limits. For instance, when comparing Qwen2-0.5B-Instruct and Qwen2-1.5B-Instruct, both trained in the same manner, the larger Qwen2-1.5B-Instruct model exhibits greater memory capacity, allowing it to retain more content.

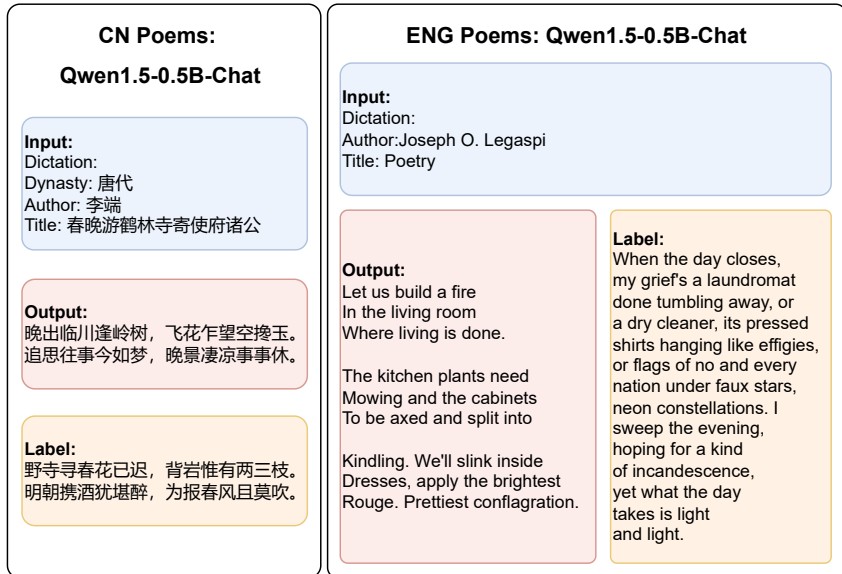

Figure 3: The examples of wrong prediction of CN Poems: Qwen1.5-0.5B-Chat and ENG Poems: Qwen1.5-0.5B-Chat which were fine-tuned separately on CN Poems and ENG Poems.

## 3.4 THE OUTPUTS LENGTH EFFECT

Additionally, we believe that the length of the output text has a significant impact on the memory capabilities of LLMs - the longer the text, the harder it is to remember. To verify this, we set the combined length of the input and output text in the CN Poems dataset to be between 256 and 512 characters. We used Chinese text because the relationship between the token length and the original text length is not fixed in English. After fine-tuning the model for 100 epochs on CN Poems, the results are shown in Table 2. It is evident that as the text length increases, the difficulty for the model to remember the content also increases.

Table 2: The memory ability of Qwen1.5-0.5B-Chat, Qwen2-0.5B-Instruct, Qwen2-1.5B-Instruct, bloom-389m-zh, bloom-1b4-zh on CN Poems in the condition of longer prediction.

| Models | | Qwen1.5 -0.5B-Chat | Qwen2 -0.5B-Instruct | Qwen2 -1.5B-Instruct | bloom -389m-zh | bloom -1b4-zh |
|---|---|---|---|---|---|---|
| CN Poems | Acc | 44.9 | 56.85 | 86.95 | 68.6 | **93.65** |

## 4 A COMPARISION BETWEEN HUMAN BRAIN AND LLMS

Based on the definition of memory in Section 3 and the experimental results, we believe that LLMs do possess memory capabilities. It's important to distinguish between LLM memory and database storage. Database storage involves keeping content on physical media (like hard drives or books) that can be searched or modified based on conditions, while LLM memory refers to the dynamic approximation of corresponding outputs using internal weights and inputs.

From the perspective of functionality, we argue that LLMs and human memory do not fundamentally differ; both can be understood as dynamically approximating results based on inputs. For example, as shown in Figure 2, LLMs can recite entire poems solely based on their titles and authors after learning. These poems are not stored in specific areas within the model; they are dynamically generated based on input. We can only determine if an LLM remembers certain information by posing questions and examining outputs; otherwise, it remains unknown. Human memory operates similarly: we can only validate our memories by answering specific questions; otherwise, assessment is impossible. For instance, if you ask someone how many poems they remember, they may struggle to provide an exact number, but they can usually recall a specific poem if prompted. Few people

consciously memorize how many poems they know, leading to a lack of corresponding output when such input arise. Therefore, we suggest that the brain functions like a model that dynamically fits outputs based on inputs, indicating that, in a sense, the mathematical model of the human brain may resemble that of a Transformer-based dynamic approximation UAT model, potentially even as a more advanced version. However, we believe their fundamental mechanisms are the same: both rely on dynamically fitting outputs based on inputs.

Due to the complexity of the brain, many operational mechanisms remain unclear, and no reasonable conclusions currently exist regarding its specific workings. Thus, we make logical assumptions and generalizations about the brain's mechanisms based on the memory process, UAT theory, and LLMs. First, we extend the concept of memory in the brain to other cognitive abilities, such as social skills, imagination, and creativity. All of these can be attributed to the ability to infer outcomes based on existing knowledge and inputs, which we collectively refer to as reasoning ability, defined as: the capacity to generate specific results based on previously learned knowledge and specific inputs, where these results are consistent with or related to that knowledge.

Based on the definition of reasoning ability, LLM memory can also be viewed as a form of reasoning. The results from Figures 2 and 3, along with the current performance of ChatGPT-4 (Achiam et al., 2023) in generating outputs based on inputs, suggest that LLMs possess reasoning capabilities. Although the predictions in Figure 3 are incorrect, they still align with linguistic conventions and somewhat correspond to the titles of the poems. This can be seen as creativity.

So why do LLMs seem to underperform in reasoning tasks? We believe there are two main factors: model size, and data quality and quantity.

- Model Size: Generally, larger LLMs tend to be more powerful. Theoretically, as demonstrated by UAT2LLMs (Wang & Li, 2024b) and UAT2Parallel (Wang & Li, 2024a), a greater model size enhances dynamic fitting capability, leading to improved performance. Performance improvements can also be observed when comparing models like Llama from 8B to 70B (Dubey et al., 2024) and Qwen2 (Yang et al., 2024a) from 0.5B to 72B-larger models consistently show better performance.

- Data Quality and Quantity: Current LLMs have significantly benefited from training on vast datasets. The larger and higher the quality of dataset, the stronger the model's performance. The performance leap from Qwen 1.5 (Bai et al., 2023) to Qwen 2 (Yang et al., 2024a) highlights that training on high-quality data yields better results. From a human learning perspective, individuals undergo decades of education from elementary school to university. Immersed in a language-rich environment from birth, humans benefit from teachers and exams that correct linguistic issues one by one. Without such learning experiences, we would struggle to develop robust language skills.

Since we propose that both LLMs and the brain function as dynamic models that fit outputs, why build such dynamic models? What are the advantages of this approach? We believe that this dynamic fitting capability gives the brain infinite possibilities. The brain doesn't need to remember everything; it only needs to focus on what is important. Imagine if a newborn's brain were preloaded with the weights of its parents; there would be no need to fit the world because most of the external environment remains constant. In such a scenario, the brain's weights would hardly ever be updated, and the person would lose creativity. However, when we are born, our brains contain almost no knowledge about the external world. It could be viewed as the weight parameters in our brains are randomly initialized, and it is precisely this randomness that gives rise to creativity. Since the knowledge of our ancestors isn't always correct, newborns learn from their predecessors and continuously interact with the world to verify and update this knowledge. This updating process involves optimizing the brain's weight parameters. Each update may be right or wrong, but with a vast number of humans exploring the world, we gradually inch closer to the truth, ultimately leading to innovation.

A great example of dynamic fitting in the brain is Henry Molaison (Scoville & Milner, 1957; Victor et al., 1961; Milner & Klein, 2015). After his hippocampus (Bliss & Collingridge, 1993; Squire, 1992; Erickson et al., 2011; Eckardt, 1980) was damaged, he could no longer form new long-term memories, though his existing memories remained intact. We believe that the hippocampus acts as a switch controlling whether the weights responsible for long-term memory in the brain can be updated. Once the hippocampus is damaged, the brain's weight parameters can no longer change,

meaning that while past inputs (before the hippocampal damage) can still produce corresponding outputs (i.e., recalling past events), the inability to update weights prevents the formation of new memories.

## 5 CONCLUSION

In this paper, we demonstrate that LLMs possess memory capabilities, which are enabled by their Transformer-based architecture. This architecture functions as a dynamic fitting UAT model, with a strong ability to adaptively fit outputs. As a result, LLMs can recall entire content based on minimal input information. Since this memory can only be confirmed when triggered by input, we refer to it as "Schrödinger's memory." Through extensive experiments, we validated that the memory mechanism of LLMs aligns with this theory. Additionally, we compared LLMs with the human brain and found that their working mechanisms are similar, as both dynamically fit outputs based on inputs.

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
