# OpenReview forum: "Schrodinger's Memory: Large Language Models"
_ICLR.cc/2025/Conference — Submitted to ICLR 2025_

### Official Review · Reviewer_sRSv · 2024-10-28

**Soundness:** 2
**Presentation:** 3
**Contribution:** 2
**Rating:** 3
**Confidence:** 4

**Summary:**

This work explores the memory capabilities of LLMs using the Universal Approximation Theorem (UAT). The authors introduce the concept of "Schrödinger's memory" - suggesting that LLMs' memory only becomes observable when queried and remains indeterminate otherwise. The paper presents experimental results comparing different models' ability to memorize Chinese and English poems and draws comparisons between LLMs and human memory mechanisms.

**Strengths:**

The paper tries to address an important and timely question about the memory mechanisms in LLMs;
The choice of structured datasets (particularly poetry) is suited for testing memory recall in LLMs and evaluating the relationship between model architecture and memory capacity;
The attempt to connect UAT with LLMs memory is interesting and provides a mathematical basis for discussing dynamic response mechanisms in LLMs;
The comparison between human and LLM memory provides insights that could bridge cognitive science and AI.

**Weaknesses:**

1.The paper claims to use UAT to explain LLMs' memory abilities, but the connection between Eq. (3) and the memory mechanism is not rigorously established. The authors need to prove how the dynamic fitting capability directly relates to memory retention. This connection might be strengthened by providing a step-by-step derivation linking Eq. (3) to specific memory processes, or designing an experiment that directly tests the relationship between dynamic fitting and memory retention.
2.While the paper develops an extensive theoretical framework using UAT in Section 2, the experiments in Section 3 do not directly validate or connect to this theory. This gap might be addressed by quantitatively relate the parameters in Eq. (3) to the observed memory performance in the experiments.
3.Section 3.3 proposes using memory ability as an objective measure of LLM capabilities but does not relate this metric to standard benchmarks. The authors should demonstrate how their memory assessment correlates with or complements existing evaluation methods.
4.The output length effect analysis in Section 3.4 only tests up to 512 characters. The authors should investigate how memory performance degrades with sequence length and compare this with theoretical predictions from their UAT framework.
5.The paper fails to examine an essential aspect of memory systems - their ability to recognize unknown information. Specifically, there are no experiments testing whether models can reliably indicate when they encounter previously unseen poems, nor is there a comparison between "memorized" vs. "hallucinated" outputs' characteristics. The paper could benefit from adding specific experimental designs to test the models' ability to distinguish between known and unknown information. This might be achieved by including a set of previously unseen poems in the test set, or evaluating how the models distinguish between known and unknown information.
6.The experiments only measure binary success (correct/incorrect recitation) without examining the deviations as metioned in Section 3.1.
7.The comparison between human and LLM memory is largely speculative and lacks scientific rigor. Many claims about human memory mechanisms are made without proper citations or empirical support. This could be improved by designing experiments to more rigorously compare human and LLM memory, perhaps drawing on established methods from cognitive psychology.

**Questions:**

1.How do you justify that the UAT format in Equation 3 specifically relates to memory rather than a general dynamic fitting argument? For instance, are there specific properties of UAT that correspond to the characteristics of memory, such as retention, retrieval, or even forgetting? How would you define these aspects mathematically within the UAT framework?
2.Which specific aspects of memory can the theoretical framework of UAT explain better than standard Transformer-based fitting capabilities?
3.Can you provide theoretical bounds or guarantees for the memory capacity of LLMs based on your UAT analysis?
4.How does the proposed memory assessment method account for different types of memory (e.g., factual vs. procedural)?
5.Memory typically involves not only retention but also changes in recall accuracy over time or with repeated exposure to new information. Did you investigate how memory in LLMs degrades or strengthens with time, additional training, or varying contexts?

---

### Official Review · Reviewer_xaT5 · 2024-11-02

**Soundness:** 1
**Presentation:** 1
**Contribution:** 1
**Rating:** 1
**Confidence:** 5

**Summary:**

This work first relates the LM architecture to the UAT, then proposes a new definition of memory in LLMs, and tests the memorization capabilities of Qwen and Bloom models on Chinese and English poems, showing that LMs can memorize poems after 100 epochs of finetuning. It conclude with a comparison of memorization between brains and language models.

**Strengths:**

The paper investigates memory in LLMs, which is timely given the current explosion of research on LMs, and is thematically aligned with ICLR.

The decision to test memorization on poetry is a step in the right direction.

**Weaknesses:**

While the experimental work is a step in the right direction, I do not believe the paper in its current form would be a good fit for ICLR.


1. The authors aim to provide a precise definition of memory, yet the definition they provide is informal, and the only other definition they compare to is one from Wikipedia (ignoring the vast literature on, e.g., dense associate memory, episodic vs working memory etc).
	1. While the goal of Section 3.1 is to define memory, the definition provided is imprecise. For instance, is memory a function from inputs to outputs? Is it a function from (inputs, outputs) to a truth value? The authors go on to use next-token prediction accuracy as their metric for memory in Section 4. I suggest the authors to formally define memory in the next version of the manuscript.

2. The work presented is quite speculative, and the tone of writing imo would be better aligned to a non-ML conference.
	1. Section 3, especially 3.1, reads to be long-winded and imprecise-- while it tries to define memory, it instead provides many examples without a formal definition of memory.
	2. Section 4, which compares brains to LMs, does not cite the relevant literature on neuroscience (except for one case study in l427).
	3. l424 "Each update may be right or wrong, but with a vast number of humans exploring the world, we gradually inch closer to the truth, ultimately leading to innovation." This line (and similar ones) should ideally be toned down or omitted for a machine learning conference.

3. The work does not engage with the vast literature on memory in neuroscience, cognitive science and machine learning. Instead, it overinterprets and over-draws parallels between neural networks and human cognition. I'm listing several examples here:
	1. l269 The connection to human cognition requires a citation.
	2. l421: It could be viewed as the weight parameters in our brains are randomly initialized...
	3. Table l207-212: Answer 2,3,4 -> Answer 1, 2, 3. I disagree that one would conclude "minor distortions, severe distortions, and memory loss" from these examples. If the answer is "I do not know", then it could be that the person never learned Newton's first law.

4. Methodology
	1. The selected poems are common ones likely found in training data, especially for the Chinese language models. Therefore, it is not guaranteed that each poem is trained on the same number of times (t=100 epochs). How do you disentangle the effects from pre-training on these poems?
	2. The memorization metric based on accuracy is not sufficiently different from Carlini et al 2022, which defines memorization as $k$-extractability, $k$ being the number of tokens in the input prompt.
	3. The connection between experiments and UAT is never made.

5. Several unsubstantiated claims about linguistics
	1. l271 "Chinese is a more complex language" is vague-- I would remove this sentence altogether or specify exactly which components of Chinese are more complex than English with citations from the linguistic typology literature.
	2. l214-215 is rather abrupt and would go better in the introduction of the section. The second sentence "Now, we believe that LLMs also exhibit memory"-- needs citation

6. Lack of engagement with the UAT / LLMs literature [1-3], which show that Transformers, in the general case, are not universal approximators-- only under certain conditions.

[1] Alberti et al. 2023. Sumformer: Universal Approximation for Efficient Transformers
https://proceedings.mlr.press/v221/alberti23a.html

[2] Kratsios et al. 2022. Universal Approximation Under Constraints is Possible with Transformers
https://openreview.net/forum?id=JGO8CvG5S9

[3] Luo et al. 2022. Your Transformer May Not be as Powerful as You Expect
(RPE based transformers aren't Universal Approximators)

**Questions:**

See weaknesses

---

> ### Author Response · Authors · 2024-11-14
> **To the reviewer**
>
> First, I would like to express my sincere gratitude for the valuable feedback provided by the reviewer. Below, I will address each of your comments in detail.

---

> ### Author Response · Authors · 2024-11-14
> **Weaknesses**
>
> #### 1. The authors aim to provide a precise definition of memory, yet the definition they provide is informal, and the only other definition they compare to is one from Wikipedia (ignoring the vast literature on, e.g., dense associate memory, episodic vs working memory etc).
>
> #### While the goal of Section 3.1 is to define memory, the definition provided is imprecise. For instance, is memory a function from inputs to outputs? Is it a function from (inputs, outputs) to a truth value? The authors go on to use next-token prediction accuracy as their metric for memory in Section 4. I suggest the authors to formally define memory in the next version of the manuscript.
>
> ### First, we should clarify that our definitions here are specifically targeted at LLMs and human memory.
>
> ### Regarding the references suggested by the reviewer, they are simply assigned different terms in specific contexts. Memory can indeed be referred to by many different terms.
>
> ### In response to the reviewer's question: "Is memory a function from inputs to outputs?" We have clearly stated in the paper that memory in LLMs involves dynamic approximation based on inputs. We provided the corresponding mathematical model in Section 2.2. I would like to ask why memory cannot be defined as a dynamic fitting function from inputs to outputs.

---

> ### Author Response · Authors · 2024-11-14
> **Weaknesses**
>
> 2. The work presented is quite speculative, and the tone of writing imo would be better aligned to a non-ML conference.
>
> 1.Section 3, especially 3.1, reads to be long-winded and imprecise-- while it tries to define memory, it instead provides many examples without a formal definition of memory.
> 2.Section 4, which compares brains to LMs, does not cite the relevant literature on neuroscience (except for one case study in l427).
> 3.l424 "Each update may be right or wrong, but with a vast number of humans exploring the world, we gradually inch closer to the truth, ultimately leading to innovation." This line (and similar ones) should ideally be toned down or omitted for a machine learning conference.
>
>
> ### First, we need to emphasize that current research on the brain has not reached definitive conclusions. Most findings are based on observed phenomena. The phenomena we present are real and observable. If the reviewer finds any of these phenomena unreasonable, please specify which ones, and we will make the necessary corrections.
>
> ### 1. We have provided a definition of memory. Could you please specify why this definition is unreasonable? We would appreciate more detailed feedback rather than a general statement of unreasonableness. Please provide an example of what you consider to be a formal definition of memory.
> ### 2. We have illustrated the issue using an empirical example. Could you please specify which parts require citations to neurological literature and what specific references are needed?
> ### 3. We will consider adopting your suggestions in the future, but here we are engaging in academic exploration. Could you please specify the exact issue with this statement? We believe it is correct, as it reflects the process of our interaction with the real world.

---

> ### Author Response · Authors · 2024-11-14
> **Weaknesses**
>
> The work does not engage with the vast literature on memory in neuroscience, cognitive science and machine learning. Instead, it overinterprets and over-draws parallels between neural networks and human cognition. I'm listing several examples here:
>
> 1. l269 The connection to human cognition requires a citation.
> 2. l421: It could be viewed as the weight parameters in our brains are randomly initialized...
> 3. Table l207-212: Answer 2,3,4 -> Answer 1, 2, 3. I disagree that one would conclude "minor distortions, severe distortions, and memory loss" from these examples. If the answer is "I do not know", then it could be that the person never learned Newton's first law.
>
> ### 1. Yes, citations to relevant literature should be included here.
> ### 2. Our hypothesis about the human brain is intended as a conjecture. Why can't we propose such a conjecture? Currently, no one can definitively explain these phenomena, so why shouldn't we offer our own explanation?
> ### 3. Yes, it could also be due to a lack of learning. However, in a sense, a lack of learning and memory loss are equivalent, as both result in the inability to map inputs to outputs.

---

> > ### Comment · Reviewer_xaT5 · 2024-11-15
> > **Engagement with the literature**
> >
> > While I found the work to be commendable in its ambition and scope, and I respect the authors for reasoning about memory from first-principles, a condition for acceptance should be correctness, an extensive literature review, and proper contextualization of the current work.
> >
> > Here is a non-extensive list I recommend for literature review:
> >
> > 1. Carlini et al, 2022: https://openreview.net/forum?id=TatRHT_1cK \
> > Although the authors criticize this paper, I would say that has a truly actionable definition of memory, which is then thoroughly tested. I would model the structure of your paper after their setup.
> >
> > 2. Neuroscience: \
> > The authors raise some questions, e.g. "does a neuron in the brain store a word or a whole sentence?", among others that I won't list, that imply a lack of due diligence with the neuroscience literature. I would recommend reading about (1) biologically plausible learning algorithms, such as Hebbian learning, spiking neural networks, etc; and also (2) concept decoding from neural signals, e.g., https://www.nature.com/articles/s41467-020-15804-w . In particular, there is no evidence in the neurolinguistics literature that single neurons store entire words and sentences; further, it is unclear what is meant by sentences like "weight parameters in our brains". Sentences like these should use the correct terminology, e.g., do you mean synaptic weights? from the neuroscience literature and cite sources; I would start by looking up "developmental plasticity".
> >
> > Overall, I strongly agree with Reviewer 1's assessment. While I don't foresee raising my score, I hope that you don't feel discouraged by the rebuttal, and I recommend these steps to improve the quality of the paper for a future resubmission:
> >
> > 1. _Writing in an academic style._ This involves tempering claims, supporting each one with a citation if possible, supporting speculative remarks with experimental evidence. Using proper terminology (e.g. removing "weight parameters in our brains"). Statements like "Each update may be right or wrong, but with a vast number of humans exploring the world, we gradually inch closer to the truth, ultimately leading to innovation." are, for instance, not appropriate for academic writing. The reason is that the goal of academic writing is to put evidence towards a claim, and sentences like these detract from this goal. To get a better idea of an appropriate style, I would read past machine learning papers.
> >
> > 2. _An extensive literature review._ Before engaging with ideas from cognitive science and neuroscience, it's important to read up on the current literature before claiming things like "no one can definitively explain these [neural] phenomena"; without acknowledging the work that has been done towards an explanation. A better lit review is crucial to see that statements like "does a single neuron store a word, or does it store an entire sentence?" don't belong in a paper related to neuroscience; questions like these are never asked in the literature because overwhelming evidence suggests it's not true.
> >
> > 3. _Better connection between theory and experiment._ As elaborated in my previous comments and by other reviewers.
> >
> > Lastly, as R1 suggested, I would suggest submitting a version of this work to a workshop or try presenting it at local symposia to get academic mentorship. A lot of paper writing is about appropriate framing and contextualization with previous work; this is what mentors are for!

---

> > > ### Author Response · Authors · 2024-11-16
> > > **Response to Engagement with the literatur**
> > >
> > > First, thank you very much for your reply.
> > >
> > > Your comments are all very constructive, and I will follow your suggestions to make the revisions. Additionally, if possible, could you recommend any workshops? Thank you very much.
> > >
> > > There is one issue I need to clarify: "Does a neuron in the brain store a word or a whole sentence?"
> > >
> > > We raise this question to emphasize that our answer is no. The brain dynamically approximates the corresponding result based on input, so the essence of memory is dynamic approximation. The same applies to other forms of thought, which are also processes of dynamic approximation.

---

> > > > ### Comment · Reviewer_xaT5 · 2024-11-16
> > > >
> > > > Thanks! There's one upcoming workshop at ACL next year, the submission deadline will probably be in late spring. There's no website yet: https://x.com/l2m2_workshop

---

### Official Review · Reviewer_atWx · 2024-11-02

**Soundness:** 1
**Presentation:** 1
**Contribution:** 1
**Rating:** 1
**Confidence:** 4

**Summary:**

The authors investigate the concept of memory for LLMs via “memory ability assessment”. To this end, they create a novel definition of memory based on an input-output relation and connect the Transformer architecture to the Universal Approximation Theorem. They claim that the weights and biases of Transformer-based LLMs can “dynamically change according to the input” as the basis of memory. Empirically, they show that fine-tuning LLMs on English and Chinese corpora of poems enables them to recite the latter based on the information about the author and title (and dynasty for Chinese poems). They conclude by comparing human brains and LLMs, discussing the model dependence on model size, data quality, and quantity.

**Strengths:**

The general idea of connecting deep theory about functions to concrete Transformer-based LLMs (for example, via the Universal Approximation Theorem) is interesting and promising.

**Weaknesses:**

Unfortunately, the article suffers from several shortcomings that I will point out section-wise:


Introduction:

Overall, the main topic of the article should be clarified. There needs to be a proper section on related work to outline and delineate current research on this topic so that readers know what the state-of-the-art is and why this paper's contributions are contributions in the first place. This would also help readers unfamiliar with the topic to navigate the article more efficiently. Also, a large portion of the article is concerned with the Transformer architecture, but the original paper is not cited.



UAT and LLMs:

Overall, the formulation of the theoretical background severely lacks precision and formalism. A significant portion of the notation has not been defined (for example, $C(I_n)$) before using it. Furthermore, the authors write (starting in line 095/096):

“[...] then a finite sum of the following form: [...] is dense in $C(I_n)$”

However, "a finite sum" can not be dense in the set of continuous functions on the hypercube (which is meant by $C(I_n)$). The authors then continue to reference the article by Wang & Li (2024b), stating (line 131):

"[...] parameters in the multi-head attention mechanism are modified dynamically in response to the input."

It is unclear whether this refers to the forward or backward process. Overall, the entire section is very confusing, with statements like (starting in line 137):

"[...] the UAT's parameters are fixed once training is completed, [...]"

"UAT" stands for Universal Approximation Theorem - what is meant by parameters?

A large portion of section 2.1 is very similar to the beginning of Section 2 of the paper by Wang & Li (2024b) (who also do not cite the original transformer paper). As an example, the error in line 102 is the same ($\theta \in \mathbb{R}$ needs to be $\theta_j \in \mathbb{R}$).

Finally, the reference "(Cybenko, 2007)" refers to the article published in 1989 (the same year Hornik et al. published their paper) - is this reference incorrect?



The Memory of LLMs:

This section discusses memory for humans and LLMs and introduces the authors' definition of memory. They criticise other works for the lack of a fundamental theoretical framework and vague definitions of memory (see line 059 and following), but the definition presented in this work seems no different in these regards. The authors should tie their definition to the introduced theory in the previous section and make the formulation more rigorous.

Overall, there is no cited work (apart from the Wikipedia definition) in Section 3.1, which seems more like a blog post than an academic article. This leaves statements like (line 168)

"The brain does not have a structure analogous to a database for storing information."

unfounded.

The datasets are attributed to "Unknown" - although the Huggingface user accounts are available via the provided links.

The authors calculate the mean average accuracy to evaluate the models' ability to recall poems. However, it is unclear when a poem counts as predicted correctly. Did the authors employ plain string matching? If so, how are newline characters and translations by spaces handled?

Furthermore, most training details are missing. For example, what were the learning rate and batch size? Based on the provided information, no experiments are reproducible (no code is available).

Regarding Table 1: What do the hyphens stand for?

The authors also claim in line 253/254 that—based on the results in Table 1—" LLMs possess memory capabilities, which align precisely with the definition of memory we established."

In my view, the authors seem to reduce memory to a capability every overfitted LLM can develop: reproducing tokens in order. Here, I use the term "overfitted" as training for 100 epochs on such a comparatively small corpus of text (2000 poems) seems excessive.

Finally, given the same input, if the LLM could reproduce the poems in reversed order, starting with the last token (or character) and ending with the first token (or character) - would this count as memory according to the provided definition and would the metric in Equation (4) reflect this?



A Comparison Between Human Brains and LLMs:

Similar to Section 3.1, the discussion about similarities and differences between human memory and memory in LLMs at the beginning of the fourth section lacks academic references and empirical evidence to support claims such as (line 373/374)

“These poems are not stored in specific areas within the model; they are dynamically generated based on input.”

and (line 395)

“Although the predictions in Figure 3 are incorrect, they still align with linguistic conventions and somewhat correspond to the titles of the poems. This can be seen as creativity.”

Some related articles discuss the case of Henry Molaison, which led to the authors hypothesising about similar dynamics for LLMs, but these are likewise unfounded. In particular, I do not see sufficient evidence for the third contribution mentioned in the introduction.

The authors likewise state in line 082 that they

“[...] conduct a comprehensive analysis of human and LLM abilities, with a focus on memory ability.”,

which I also do not see provided.



Overall, the article severely lacks formalism, experimental details and references/experiments to support the author's claims. The introduced definition of memory needs to be improved, that is, sharpened, theoretically motivated and empirically justified.







Minor:

Grammar and spelling mistakes need to be corrected, for example:

Line 137: “This ability enables the Transformer to adaptively fit based on the input [...]”

Line 219: “We select the poems from datasets and the requirement is the combined length of the input and output to a maximum of 256 characters.”

Line 241/242: “[...] are the prediction and ground true of the i-th exmple.”

Line 351/352: “After fine-tuning the model for 100 epochs on CN Poems, the results are shown in Table 2.”

Line 406/407: “The larger and higher the quality of dataset, the stronger [...]”

**Questions:**

See Weaknesses.

---

> ### Author Response · Authors · 2024-11-13
> **To the reviewer**
>
> We sincerely appreciate the time you have taken to review our paper and for providing many valuable comments. We will address each of your suggestions in detail below.

---

> ### Author Response · Authors · 2024-11-13
> **Weaknesses**
>
> #### Overall, the main topic of the article should be clarified. There needs to be a proper section on related work to outline and delineate current research on this topic so that readers know what the state-of-the-art is and why this paper's contributions are contributions in the first place. This would also help readers unfamiliar with the topic to navigate the article more efficiently. Also, a large portion of the article is concerned with the Transformer architecture, but the original paper is not cited.
>
> ### We have conducted extensive research on the literature regarding memory in LLMs, but beyond the papers already cited, we have not found additional studies in this area. If the reviewer is aware of any relevant articles, we would be more than willing to include them in our references.
>
> ### Additionally, you are correct; we should indeed cite the original Transformer paper.

---

> ### Author Response · Authors · 2024-11-13
> **Weaknesses**
>
> #### However, "a finite sum" can not be dense in the set of continuous functions on the hypercube
>
> ### I think reviewer is wrong. This has been proofed in the paper [1].
>
> ### Kurt Hornik, Maxwell B. Stinchcombe, and Halbert L. 532 White. Multilayer feedforward networks are universal approximators. Neural Networks, 2:359–366, 1989. 2, 7

---

> > ### Comment · Reviewer_atWx · 2024-11-21
> >
> > Note that "a finite sum" is a single point in the set of continuous functions on the hypercube. It cannot be dense in the topological sense. You need to reframe everything in terms of (sub)sets. Definition 2.6 in the paper of Hornik et al. gives a precise definition of denseness. This is just an example of the lack of formalism in the paper, which is my primary criticism here.
> >
> > However, I listed several more questions and points of critique in my review and would be grateful if you could address them.

---

### Official Review · Reviewer_cmqq · 2024-11-04

**Soundness:** 1
**Presentation:** 2
**Contribution:** 1
**Rating:** 1
**Confidence:** 4

**Summary:**

The authors study the concept of memory in LLMs. They draw inspiration from recent work demonstrating how UAT can be applied to Transformer architectures, propose a new definition of memory that can be applied to both humans and transformer LLMs, and introduce a concept of "Schrodinger's Memory" - a type of memory that can only be detected upon probing. Lastly, the authors run a few experiments, demonstrating remarkable memory capacity of transformer architectures and proving conclusively that transformers are capable of a certain form memory.

**Strengths:**

The paper addresses an important topic. It is crucial that we study in detail the similarities and differences in memory processes between humans and LLMs, as well as devote plenty of effort do understand how memory manifests in transformer architectures.

The paper is relatively clearly written.

The work has both a theoretical and a practical component. It is always commendable when practical experiments are informed by theory.

**Weaknesses:**

** Novelty **
Unfortunately, the paper is not sufficiently novel to pass the high standards of the ICLR conference. The main empirical result in the paper is a rediscovery of a well-known phenomenon of overfitting. As is well-known, LLMs do indeed possess a remarkable ability for verbatim memorization of items in the training set.

**Quality**
Unfortunately, the quality of the literature review or the theoretical justification of the proposed work is not sufficient to pass the high standards of the ICLR conference. The authors pose a number of rhetorical questions, such as "So, is this sentence stored within a single neuron?" entirely ignoring decades of memory research in Psychology, Cognitive Science, and Neuroscience. It is also staggering to see them claim 'in summary, the term ”memory” was traditionally used to refer specifically to human memory before the emergence of LLMs'. This entirely ignores the research on memory in animals and even plants, not to mention the related concepts of the memory of materials, cultural memory, and a plethora of other related concepts.

I find it commendable that the authors ask ambitious, fundamental questions, such as "moreover, if this memory is stored in a fixed set of neurons, then every time the question is raised, the response should be identical, since the retrieval would be from the same static content." Unfortunately, however, such questions have been extensively studied not even for centuries, but for millennia. If these questions are truly of interest, I suggest starting with the works of Pavlov, Skinner, and Tolman to gain a historical perspective on how this and related questions have been approached by the scientific community.

**Questions:**

I struggle to understand why the authors introduced the concept of Schrodinger's Memory. It seems extremely broad. Isn't human memory also an example of Schrodinger's Memory? We have no other way of finding out whether a human has a specific memory other than presenting certain stimuli and recording reactions.

Moreover, some aspects of the Schrodinger's experiment analogy don't work: for example, the memory is not in an indeterminate state before querying. Moreover, the act of querying (observing) does not affect the memory in the same way as the wave function is affected in quantum experiments.

Suggestions:
I deeply hope that my negative review does not prevent the authors from further pursuing research in this direction. I highly suggest taking a step back and focusing on a more narrow aspect of LLM memory. It is also absolutely crucial to perform a thorough, deep literature analysis before performing any experiments. For example, the authors currently ask a lot of question about human memory organization in the brain, but they never mention an enormous body of literature that has been written on the topic. Similarly, the main result they obtained is a well-known phenomenon called "training set overfitting". It might be difficult to navigate the many unspoken rules of academic research, hence I highly suggest that the authors seek input from members of scientific community - mentorship arrangements, internships, and other forms of peer guidance help authors create a better and refined version of their research.

**Details Of Ethics Concerns:**

I do not claim that the paper was written by an LLM, but it might have been. It has a number of internal inconsistencies that I can't easily explain.

For example: "we provide a brief overview of the UAT, which was first proposed by Cybenko (2007)". Then, in the next paragraph:  "Hornik et al. (1989) further demonstrates that multilayer feedforward networks conform to the UAT".

Maybe it's just a citation year issue, e.g. they are citing a reprint, but I just can't understand how this text did not raise the authors' eyebrows and was not corrected.

---

> ### Author Response · Authors · 2024-11-13
> **To reviewer**
>
> We would like to express our sincere gratitude to the reviewer for taking the time to review this paper. We deeply appreciate your effort and valuable feedback. Below, we address the questions you raised and look forward to receiving your further suggestions.

---

> ### Author Response · Authors · 2024-11-13
> **Weaknesses**
>
> #### ** Novelty ** Unfortunately, the paper is not sufficiently novel to pass the high standards of the ICLR conference. The main empirical result in the paper is a rediscovery of a well-known phenomenon of overfitting. As is well-known, LLMs do indeed possess a remarkable ability for verbatim memorization of items in the training set.
>
> ### First, this paper is grounded in a clear theoretical foundation, as outlined in Section 2. We explicitly state that the essence of Transformer-based LLMs is dynamic approximation based on the input. We would like to understand why the well-known issue of overfitting cannot be considered a point of discussion. Additionally, we believe that memory, to some extent, can be viewed as overfitting. Could the reviewer kindly clarify why memory should not be considered overfitting?

---

> > ### Comment · Reviewer_cmqq · 2024-11-14
> > **Replied in another comment**
> >
> > See my reply "About a mathematical theory of memory" where I give a more general reply.
> >
> > Briefly, on this issue specifically: overfitting can be seen as a type of memory. But
> > a) The opposite is not true. Memory is more than overfitting, we can generalize when we retrieve information from memory, while overfitting, by definition, is associated with poor generalization.
> > b) Regardless of the relationship between memory and overfitting, running a new experiment to show that LLMs can overfit is neither novel nor valuable (it's an extremely well-known fact already).

---

> > > ### Author Response · Authors · 2024-11-14
> > > **Replied in another comment**
> > >
> > > #### See my reply "About a mathematical theory of memory" where I give a more general reply.
> > >
> > > #### Briefly, on this issue specifically: overfitting can be seen as a type of memory. But a) The opposite is not true. Memory is more than overfitting, we can generalize when we retrieve information from memory, while overfitting, by definition, is associated with poor generalization. b) Regardless of the relationship between memory and overfitting, running a new experiment to show that LLMs can overfit is neither novel nor valuable (it's an extremely well-known fact already).
> > >
> > > **A:**  a) First, we never claimed that memory is necessarily overfitting. Instead, we use the overfitting phenomenon in LLMs to illustrate the concept of memory. Second, to some extent  our brains are overfitting to the real world we encounter, and the resulting phenomenon is generalization.
> > >
> > > For example, why can't an expert in mathematics generalize from one or two examples to the entire field of physics? While this example is somewhat extreme, a simpler one illustrates the point better: why can't a long-distance runner with excellent jumping ability achieve the same jumping standards as a professional jumper after just one or two training sessions? The reason is that they need to overfit to the specific task.
> > >
> > > The key point is that because we overfit to the real world, the result appears as generalization.
> > >
> > > **A:**  b) Our goal has never been to overfit; rather, our goal is to explain the phenomenon of memory.

---

> ### Author Response · Authors · 2024-11-13
> **Weaknesses**
>
> #### Quality Unfortunately, the quality of the literature review or the theoretical justification of the proposed work is not sufficient to pass the high standards of the ICLR conference. The authors pose a number of rhetorical questions, such as "So, is this sentence stored within a single neuron?" entirely ignoring decades of memory research in Psychology, Cognitive Science, and Neuroscience. It is also staggering to see them claim 'in summary, the term ”memory” was traditionally used to refer specifically to human memory before the emergence of LLMs'. This entirely ignores the research on memory in animals and even plants, not to mention the related concepts of the memory of materials, cultural memory, and a plethora of other related concepts.
>
> ### We would like to emphasize that the focus of this study is on LLMs, with the primary subject of discussion being humans or animals. Our fundamental goal is to explore the issue of memory in humans or LLMs from a specific mathematical theoretical perspective. Regarding your comment: *"So, is this sentence stored within a single neuron?“,  which seems to overlook decades of memory research in psychology, cognitive science, and neuroscience*. Could you kindly point to specific literature that provides concrete mathematical conclusions on this matter? To the best of our knowledge, such references do not exist, but we would be happy to cite any relevant sources you can provide. We want to highlight that this paper presents specific mathematical derivations and, in conjunction with the well-known case of Henry Molaison's hippocampal damage and memory issues, offers a reasonable explanation.

---

> > ### Comment · Reviewer_cmqq · 2024-11-14
> > **Sources**
> >
> > I provided sources on mathematical modeling in Cognitive Science in another reply. The question of how memory is stored in human brain is not a mathematical question, but an empirical one. I don't see any value of me listing hundreds of possible sources here. If the authors are genuinely interested in the subject, I suggest starting with the concept of "grandmother cell" (https://en.wikipedia.org/wiki/Grandmother_cell) or "memory" more generally (https://en.wikipedia.org/wiki/Memory). There are numerous sources on the matter linked in the corresponding Wikipedia pages.

---

> > > ### Author Response · Authors · 2024-11-14
> > > **Sources**
> > >
> > > I provided sources on mathematical modeling in Cognitive Science in another reply. The question of how memory is stored in human brain is not a mathematical question, but an empirical one. I don't see any value of me listing hundreds of possible sources here. If the authors are genuinely interested in the subject, I suggest starting with the concept of "grandmother cell" (https://en.wikipedia.org/wiki/Grandmother_cell) or "memory" more generally (https://en.wikipedia.org/wiki/Memory). There are numerous sources on the matter linked in the corresponding Wikipedia pages.
> > >
> > > **A:**First, I would like to point out that this aligns with what I mentioned earlier: communication of thoughts between individuals can be quite challenging. You view memory as an empirical question, while I believe that memory can fundamentally be expressed mathematically.
> > >
> > > Let me explain why I hold this view:
> > >
> > > First, the initial neural network can be considered a perceptron, which was designed based on human neurons. However, due to subsequent changes in form, it is no longer seen as having a direct relationship with the brain. In my work, I reconnect deep learning with the brain, and therefore, I believe that whether considering the original design principles of neural networks or the phenomena exhibited by modern LLMs, they can be regarded as simulations of the brain.

---

> > > > ### Comment · Reviewer_cmqq · 2024-11-27
> > > > **re: Rebuttal**
> > > >
> > > > I have read other reviewers and the authors' replies. At present, unfortunately, I can not recommend acceptance and the score remains unchanged.
> > > >
> > > > That being said, I appreciate the authors' engagement with the matter and I hope that my feedback and that of other reviewers proves helpful in the future. The question of how memory is represented in the brain and in LLMs is fascinating and extremely deep. The desire to unite mathematical, biological, and machine-learning based approaches is natural and important. I truly hope that the authors find a way to narrow down and re-focus their work & publish it in the future.
> > > >
> > > > P.S. Another great place to start on the topic of mathematical modeling of cognition, especially in the context of neural-networks, is this classic book, made freely available by its authors: https://stanford.edu/~jlmcc/papers/PDP/

---

> ### Author Response · Authors · 2024-11-13
> **Questions**
>
> #### Suggestions: I deeply hope that my negative review does not prevent the authors from further pursuing research in this direction. I highly suggest taking a step back and focusing on a more narrow aspect of LLM memory. It is also absolutely crucial to perform a thorough, deep literature analysis before performing any experiments. For example, the authors currently ask a lot of question about human memory organization in the brain, but they never mention an enormous body of literature that has been written on the topic. Similarly, the main result they obtained is a well-known phenomenon called "training set overfitting". It might be difficult to navigate the many unspoken rules of academic research, hence I highly suggest that the authors seek input from members of scientific community - mentorship arrangements, internships, and other forms of peer guidance help authors create a better and refined version of their research.
>
> ### After line 472, we cite numerous articles on memory, as well as specific cases related to human memory.

---

> > ### Comment · Reviewer_cmqq · 2024-11-14
> > **On human memory discussion**
> >
> > Line "472" refers to a middle of the author list in an OpenAI paper in your citations. Line 427 offers a short paragraph near the very end of your paper about the role of hippocampus in human memory. It is a vastly insufficient discussion for something that is a central theme of the paper. It also contradicts the author's earlier claim that "the brain does not have a structure analogous to a database for storing information," because in some ways, hippocampus can be seen as analogous to such a database (though of course it's not a perfect analogy).

---

> > > ### Author Response · Authors · 2024-11-14
> > > **On human memory discussion**
> > >
> > > #### Line "472" refers to a middle of the author list in an OpenAI paper in your citations. Line 427 offers a short paragraph near the very end of your paper about the role of hippocampus in human memory. It is a vastly insufficient discussion for something that is a central theme of the paper. It also contradicts the author's earlier claim that "the brain does not have a structure analogous to a database for storing information," because in some ways, hippocampus can be seen as analogous to such a database (though of course it's not a perfect analogy).
> > >
> > > **A:** I believe there might be a misunderstanding. In my paper, I clearly state that the hippocampus acts as a switch for long-term memory, not as a database. If it were a database, individuals with hippocampal damage would lose their long-term memories entirely. However, he still retains his long-term memories, which supports my argument. Therefore, I believe the case I presented is valid.

---

> ### Author Response · Authors · 2024-11-13
> **Questions**
>
> #### I struggle to understand why the authors introduced the concept of Schrodinger's Memory. It seems extremely broad. Isn't human memory also an example of Schrodinger's Memory? We have no other way of finding out whether a human has a specific memory other than presenting certain stimuli and recording reactions.
>
> #### Moreover, some aspects of the Schrodinger's experiment analogy don't work: for example, the memory is not in an indeterminate state before querying. Moreover, the act of querying (observing) does not affect the memory in the same way as the wave function is affected in quantum experiments.
>
> ### Here, when we refer to Schrödinger's memory, we mean that the existence of memory can only be determined after a specific stimulus, and until then, it remains uncertain. This is not to suggest that memory is a superposition state as in quantum mechanics.
>
> ### Moreover, memory is in an uncertain state for both external observers and the individual who possesses the memory, because it can only be confirmed after a specific stimulus. For example, if I don’t ask you about your first experience eating an apple, would you recall it during the review process? If you don’t recall it, how can you determine whether you have that memory?

---

> ### Author Response · Authors · 2024-11-13
> **Quality**
>
> #### I find it commendable that the authors ask ambitious, fundamental questions, such as "moreover, if this memory is stored in a fixed set of neurons, then every time the question is raised, the response should be identical, since the retrieval would be from the same static content." Unfortunately, however, such questions have been extensively studied not even for centuries, but for millennia. If these questions are truly of interest, I suggest starting with the works of Pavlov, Skinner, and Tolman to gain a historical perspective on how this and related questions have been approached by the scientific community.
>
> ### What we are describing here is a practical phenomenon that should exist, and in Section 2, we provide the specific theoretical framework for dynamic approximation in memory mechanism. However, I did not find a concrete mathematical theory in the content you provided. We need to emphasize that we should focus on the mathematical nature of memory, rather than just the experimental phenomena of memory. If you have any questions regarding memory, we are happy to address them from the perspective of this theory.

---

> > ### Comment · Reviewer_cmqq · 2024-11-14
> > **About a mathematical theory of memory**
> >
> > It is possible to provide a mathematical description of anything, it does not make the description valuable. The mathematical description needs to predict new, unexpected empirical data to be valuable (think Einstein's prediction of Mercury's perihelion), or to simplify the description of some known models/observations (e.g. Newton's first-principles explanations of planetary motion). Unfortunately, your paper only provides a new way to formalize already known phenomena, but it does not offer and meaningful simplifications or insightful predictions.
> >
> > Overall, I believe that a lot of the questions I received from the authors stem from a fundamental misunderstanding of scientific methodology & the relationship between mathematical theorizing/modeling and empirical studies.
> >
> > I still highly suggest turning to the scientific community for pee-to-peer guidance. Alternatively/additinally, I highly suggest "Computational Modeling in Cognition: Principles and Practice" by S.Lewandowsky and S.Farrell, which would be a good foundational text on the matter, that, if studied in depth, will provide a good foundation for both the role and the use of models in Cognitive Science, and will give a good overview of existing mathematical models of various cognitive processes (including memory).
> >
> > I hope that the authors take the unanimous and, at present, negative feedback from the reviewers as a learning opportunity and try to improve their approach in the future. As of now, unfortunately, I see little evidence of that. The (short) time between when the reviews became visible and the author replies shows that they did not meaningfully engage with the references I provided. Their claim "I did not find a concrete mathematical theory in the content you provided" means exactly what it states - they did not find it; it does not mean that there are no mathematical theories of memory psychology, neuroscience, or Cognitive Science. To clarify: the sources I provided were given as a starting point in a more comprehensive literature review, not an exhaustive list.

---

> > > ### Author Response · Authors · 2024-11-14
> > > **About a mathematical theory of memory**
> > >
> > > #### It is possible to provide a mathematical description of anything, it does not make the description valuable. The mathematical description needs to predict new, unexpected empirical data to be valuable (think Einstein's prediction of Mercury's perihelion), or to simplify the description of some known models/observations (e.g. Newton's first-principles explanations of planetary motion). Unfortunately, your paper only provides a new way to formalize already known phenomena, but it does not offer and meaningful simplifications or insightful predictions.
> > >
> > > First, thank you for your guidance. I have learned a lot from this review, your comments and those of many other reviewers are very valuable, and I will correct them later.
> > >
> > > 1. To begin, since the issue is complex, I’ll first present a viewpoint and then address your question. I’m not certain if my explanation below will clearly convey my thoughts, but **I want to emphasize that this is purely an academic discussion, with no intention whatsoever of contradicting your perspective.**
> > >
> > > 2.  **Viewpoint 1**: The human brain can be considered a large, dynamic fitting function, continually adjusting its internal "weights" through interaction with the real world. In this process, some of these adaptations are accurate, while others are not. However, since these adjustments are grounded in real-world experience, whether correct or incorrect, they remain bounded by the constraints of reality. For instance, people cannot imagine something they have never encountered in some form. Take Picasso, for example: while his work is regarded as innovative, it’s possible that it simply stemmed from his exposure to distorted perspectives through mirrors or other natural distortions, which he then exaggerated and developed into his distinctive style. Yet fundamentally, his work did not transcend the bounds of his experiences with reality.
> > >
> > >  **Viewpoint 2**: It’s important to emphasize that the second viewpoint also describes the way the brain processes thoughts; it’s absolutely not directed at you personally, as I am one of those who experiences this phenomenon too. Because our brains learn based on interactions with the real world, we develop our own "fixed weights" through extensive learning, creating an equation like *y = f(x)*. This means that if others have a different perspective or produce a different output from the same input, we tend to reject their viewpoint. This is also why it’s often difficult to persuade others to see things our way.
> > >
> > > **Anwers**： When you refer to the “description of some known models/observations,” why can’t this mathematical model serve as a representation or simplification of the brain itself? In many ways, LLMs already seem remarkably close to human cognitive processes, and we have here provided a mathematical description alongside a comparison to memory phenomena in the brain.

---

> ### Author Response · Authors · 2024-11-13
> **Flag For Ethics Review**
>
> I would like to understand the reason why I couldn't write it this way. Could you kindly explain? Thank you so much.

---

> > ### Comment · Reviewer_cmqq · 2024-11-14
> > **Clarification**
> >
> > It was confusing because when you cite a publication from 2007 as the work that introduced UAT, while in the next sentence you refer to another work about UAT from 1989.

---

> ### Author Response · Authors · 2024-11-14
> **Clarification**
>
> First, I would like to express my sincere gratitude to the reviewer for the prompt response.
>
> Regarding this issue, it appears that the citation I used may have caused some confusion. The article in question seems to have been updated after my initial reference, which led to an inconsistency in the publication year. Alternatively, there may have been an error in my citation source itself. In either case, it is indeed the same article, and I have made the necessary corrections. Thank you very much for bringing this to my attention.

---

### Author Response · Authors · 2024-11-28

Thank you very much for the efforts and valuable feedback provided by all the reviewers. I will carefully follow your suggestions and make the necessary revisions in the subsequent modifications.

However, I would like to express that, in my opinion, the human brain is indeed a dynamic fitting model. Otherwise, many activities of the human brain would be difficult to explain. If we consider the brain as a dynamic model, many phenomena become more understandable. However, due to the current state of brain research, there are yet no systematic and conclusive results available.

Thank you again for your support.

---

### Meta-Review · Area_Chair_44Ty · 2024-12-08

**Metareview:**

This paper examines memory capabilities in large language models, proposing a new definition for memory and testing LLMs' poetry memorization abilities. All reviewers recommend rejection due to insurmountable issues with theoretical rigor, experimental methodology, and literature engagement.

**Additional Comments On Reviewer Discussion:**

All reviewers were in consensus about the above assessment.

---

### Decision · Program_Chairs · 2025-01-22

Reject